# Role of optimization algorithms based fuzzy controller in achieving induction motor performance enhancement

M. A. Hannan [1✉], Jamal Abd. Ali[2], M. S. Hossain Lipu[3✉], A. Mohamed[3], Pin Jern Ker [1],
T. M. Indra Mahlia [4], M. Mansor[1], Aini Hussain [3], Kashem M. Muttaqi [5] & Z. Y. Dong [6]

Three-phase induction motors (TIMs) are widely used for machines in industrial operations. As an accurate and robust controller, fuzzy logic controller (FLC) is crucial in designing TIMs control systems. The performance of FLC highly depends on the membership function (MF) variables, which are evaluated by heuristic approaches, leading to a high processing time. To address these issues, optimisation algorithms for TIMs have received increasing interest among researchers and industrialists. Here, we present an advanced and efficient quantum-inspired lightning search algorithm (QLSA) to avoid exhaustive conventional heuristic procedures when obtaining MFs. The accuracy of the QLSA based FLC (QLSAF) speed control is superior to other controllers in terms of transient response, damping capability and minimisation of statistical errors under diverse speeds and loads. The performance of the proposed QLSAF speed controller is validated through experiments. Test results under different conditions show consistent speed responses and stator currents with the simulation results.

[1] Department of Electrical Power Engineering, College of Engineering, Universiti Tenaga Nasional, Kajang 43000, Malaysia. [2] General Company of Electricity Production Middle Region, Ministry of Electricity, Baghdad 10001, Iraq. [3] Department of Electrical, Electronic and Systems Engineering, Universiti Kebangsaan Malaysia, Bangi 43600, Malaysia. [4] School of Information, Systems and Modelling, University of Technology Sydney, Ultimo, NSW 2007, Australia. [5] School of Electrical, Computer and Telecommunications Engineering, University of Wollongong, Wollongong, NSW 2522, Australia. [6] School of Electrical Engineering and Telecommunications, UNSW, Kensington, NSW 2033, Australia. ✉email: hannan@uniten.edu.my; lipu@ukm.edu.my

Three-phase induction motor (TIM) is considered a high energy consuming appliance used in industrial and commercial applications[1–3]. TIMs account for ~60% of total electricity consumption for electrical to mechanical transformation of energy[4,5]. High reliability, simple design, ruggedness, low cost and ease of maintenance are the main advantages of TIM[6,7]. However, the dynamic configuration of TIMs is a nonlinear system that cannot be easily explained from a theoretical point of view because of rapid changes in load or speed[8–10]. Therefore, an advanced and robust controller is required to enhance the strength and performance of TIM[11,12]. The scalar control (i.e., V/F control) method has been the commonly used technique to achieve reasonable speed under differ load settings of TIM. The scalar control exhibits simple design, structure and low price. Moreover, this method does not consider the parameters of motors and can control medium to the high speed of TIM effectively[13].

The conventional controller, namely, proportional–integral–derivative (PID) has been widely applied to adjust the main parameters of TIM, including rotor flux, torque, speed, current and voltage[14,15]. However, PID has shortcomings in terms of appropriate parameter selection due to the trial-and-error (TE) considerations. The artificial intelligence (AI) based controllers including artificial neural network and adaptive neuro-fuzzy inference systems have been performing satisfactorily in motor applications such as fault identification[16], speed assessment[17] and harmonics and torque ripple minimization[18]. However, the AI-based controllers have drawbacks concerning huge data requirement, long learning and training duration. Fuzzy logic controller (FLC) is extensively utilised in real-time TIM control using adaptive modelling under sudden[19,20]. Furthermore, FLC can operate in highly linear and nonlinear systems without considering any mathematical model[21,22]. Nevertheless, the accuracy of FLC depends on the suitable design and the optimal number of membership functions (MFs), as well as appropriate fuzzy rule generation[23]. Generally, a TE procedure is used to determine these variables; however, this procedure causes a substantial delay in control operation[24].

At present, the role of optimisation techniques in industrial applications has attracted massive attention because of their high accuracy, efficiency and adaptability that provides high-quality results[25–27]. Optimisation techniques have been highly explored in FLC based TIM drives for the appropriate tuning of control parameters that results in high performance and efficiency[28,29]. Ali et al.[30] introduced backtracking search algorithm (BSA) based FLC for controlling an induction motor speed, thus avoiding exhaustive traditional TE procedure for obtaining MFs. Ranjani & Murugesan[31] proposed particle swarm optimization (PSO) based FLC to determine the optimal fuzzy parameters for achieving the minimum value of the objective function (OF). Pan et al.[32] developed an optimal FLC utilizing genetic algorithm (GA) and PSO through the adjustment of control parameters to minimize the OF. Shareef et al.[33] established lighting search algorithm (LSA) based FLC to overcome the TE process in achieving the suitable value of MFs. Mutlag et al.[34] designed an advanced controller using differential search optimization based FLC to obtain the lowest value of OF and best value of MFs. Ochoa et al.[35] deployed Type-1 and Interval Type-2 fuzzy systems to enhance the performance of differential evolution (DE) algorithm to achieve dynamic adaptation of the mutation parameters as well as optimize the MFs. Castillo et al.[36] analyzed and compared the FLC optimization algorithms including bee colony optimization (BCO), DE, and harmony search algorithms. Melin et al.[37] applied shadowed type-2 fuzzy MFs to reduce the computational cost in control applications. Castillo et al.[38] optimized the generalized type-2 fuzzy logic system with BCO to achieve the optimal configuration of MFs. However, heuristic optimisation techniques exhibit performance variation because of the size and population of their dimension problem in each system[39]. Moreover, some methods show unequal global and local searching abilities in obtaining optimal results in search space[23]. To overcome these challenges, numerous studies have focused on improving search performance through quantum mechanics theories applied in optimisation[40–42].

The execution of TIM drive through the experimental platform is carried out using dSPACE, field-programmable gate array (FPGA), or digital signal processor (DSP). The dSPACE and FPGA have illustrated effectiveness in the implementation of grid-integrated voltage source inverter[43] and five-phase voltage source inverter[44], respectively. Nevertheless, dSPACE and FPGA have shortcomings in terms of cost and working method that cannot operate on a standalone basis. In contract, DSP offers benefits with regard to cost-effectiveness, low power consumption, fast computational capability, and embedding processor[45,46] and has been excellent in operating TIM drive[47] and permanent magnet synchronous motor[48].

In this study, we propose quantum-inspired lightning search algorithm (QLSA) to avoid the exhaustive conventional heuristic technique in obtaining the suitable value of the MFs. We apply the QLSA to a group of fourteen benchmark functions and compare with other optimisation techniques by using different benchmark functions. We present an optimal QLSA-based FLC (QLSAF) speed controller to tune and minimise the OF under different speed and load conditions. We implement the prototype of the QLSAF speed controller using V/f control with pulse width modulation switching technique and low-cost single-chip DSP-TMS320F28335 control board. We validate the proposed method by experiments and compare with the simulation results. The results validate and confirm the implementation of the proposed algorithm in a multi-induction motor drive.

## Results

**QLSA performance evaluation**. The accuracy, adaptability and efficiency of QLSA are assessed using the 14th benchmark functions towards obtaining the global minimum value. The results are presented in the box plot and compared with the LSA, BSA, gravitational search algorithm (GSA) and PSO (Fig. 1). Details of the comparative optimisation algorithms are depicted in the Supplementary File. The accuracy of QLSA is nearly adjacent to the global minimum in group 1 benchmark functions for Sphere (F1), Step (F2) and Quartic (F3). The second test is implemented using group 2 benchmark functions, and results indicate that the QLSA reaches the best global minimum for Schwefel 2.22 (F4), Schwefel 1.2 (F5), Schwefel 2.21 (F6) and Rosenbrock (F7). QLSA is also verified under group 3 benchmark functions, where the complexity level of the optimisation problem increases. QLSA reaches the best global minimum for F8 and the near-global minimum for F9 and F10. These results demonstrate the strong computational capacity of QLSA in obtaining any local minimum. The proposed QLSA is tested through the benchmark functions of group 4 including F11, F12, F13 and F14 (Supplementary Fig. 6). The results illustrate that the best global minimum for QLSA is found in F11 and F12, and the near-global minimum is achieved in other functions. In summary, the results shown in the box plot demonstrate that QLSA performs satisfactorily in most of the tested functions (Supplementary Tables 3–6). The results are further elaborated using convergence characteristic curves (Supplementary Fig. 7). QLSA reaches the global minimum rapidly in comparison with the other optimisation methods. Thus, the proposed algorithm exhibits excellent convergence characteristics under different function optimisations Fig. 2.

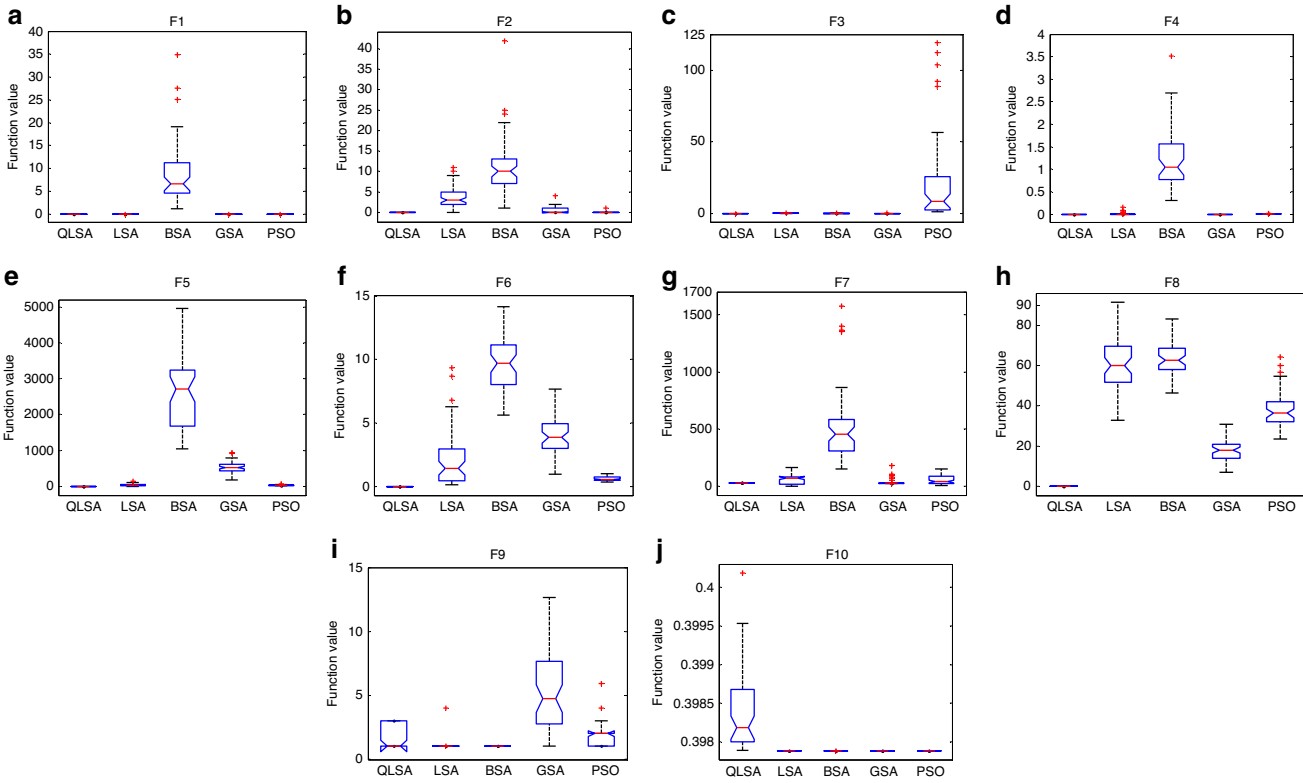

**Fig. 1 The global optimization performance assessment of QLSA, LSA, BSA, GSA and PSO under different benchmark functions. a** The global optimisation results for QLSA, LSA, BSA, GSA and PSO in benchmark function F1 (Sphere) is obtained based on dimension problem, search space and function minimum (Supplementary Table 1). **b** The global optimisation results for QLSA, LSA, BSA, GSA and PSO in benchmark function F2 (Step) is obtained based on dimension problem, search space and function minimum (Supplementary Table 1). **c** The global optimisation results for QLSA, LSA, BSA, GSA and PSO in benchmark function F3 (Quartic). **d** The global optimisation results for QLSA, LSA, BSA, GSA and PSO in benchmark function F4 (Schwefel 2.22). **e** The global optimisation results for QLSA, LSA, BSA, GSA and PSO in benchmark function F5 (Schwefel 1.2). **f** The global optimisation results for QLSA, LSA, BSA, GSA and PSO in benchmark function F6 (Schwefel 2.21) is obtained based on dimension problem, search space and function minimum (Supplementary Table 1). **g** The global optimisation results for QLSA, LSA, BSA, GSA and PSO in benchmark function F7 (Rosenbrock). **h** The global optimisation results for QLSA, LSA, BSA, GSA and PSO in benchmark function F8 (Rastrigin). **i** The global optimisation results for QLSA, LSA, BSA, GSA and PSO in benchmark function F9 (Foxholes). **j** The global optimisation results for QLSA, LSA, BSA, GSA and PSO in benchmark function F10 (Branin) is obtained based on dimension problem, search space and function minimum (Supplementary Table 1).

**Simulation results of optimal control system in TIM**. The QLSAF controller is designed and implemented under MATLAB/ Simulink environment. To verify the effectiveness of QLSAF, the results are compared with various fuzzy speed controllers, including LSA-based fuzzy (LSAF), BSA-based fuzzy (BSAF), GSA-based fuzzy (GSAF) and PSO based fuzzy (PSOF) speed controllers. The accuracy of the proposed QLSAF controller is tested under three test cases, namely, sudden changes in step response, down-to-up-to-down (DTUTD) step SR and ramp response (RS)[49]. The convergence characteristic curves are generated by the different optimised controllers that illustrate the OF (Fig. 3a). The results demonstrate that the QLSAF speed controller rapidly responds towards obtaining the lowest value of the OF in comparison with the other optimal controller methods. In this research, the maximum border and change of errors for MFs is between −3 and 3, and the output of MFs is located between −6 and 6. The QLSAF optimisation technique is used to determine the optimal MF values between the maximum borders of each TIM. The optimised values of MFs for error, change of error and output from the QLSAF speed controller are depicted in Fig. 3b–d, respectively. A 3D diagram (Fig. 3e) indicates the relationship between the inputs (error and change of error) and the output (slip speed).

A step response test is performed to assess the adaptability of the QLSAF controller under rapid variation in speed response

(SR) and load difference. The step SR consists of three cases for each TIM, namely, sudden change from three quarters to full speed, half speed to full speed and quarter speed to full speed with no-load and load conditions. The SR of the motor increases from 105 rad/s to 140 rad/s at 0.3 s, and then the speed drops to 105 rad/s from 140 rad/s at 0.6 s without loading, as shown in Fig. 4a. Accordingly, the peak stator currents (SCs) increase from 0.6 A with 37.5 Hz to 0.65 A with 50 Hz. Similar speed variation is further applied under 2 Nm load conditions, thereby increasing the peak SCs from 1.05 A to 1.15 A (Fig. 4b). The mean absolute error (MAE), root mean squared error (RMSE) and standard deviation (SD) are 3.4720%, 16.5700% and 16.2653%, respectively (Supplementary Table 7). In Fig. 4c, a change in speed is observed from 70 rad/s to 140 rad/s at 0.3 s and then from 140 rad/s to 70 rad/s at 0.6 s under the variation in SCs from 0.55 A with 25 Hz to 0.65 A with 50 Hz at no-load condition. In Fig. 4d, the SCs change from 0.95 A to 1.15 A under 2 Nm load condition; the figure shows SRs that are similar to those in Fig. 4c. The MAE, RMSE and SD of QLSAF are 2.1009%, 10.4828% and 10.2860%, respectively (Supplementary Table 7). In Fig. 4e, the SR initially increases from 35 rad/s to 140 rad/s at 0.3 s and then declines to 30 rad/s from 140 rad/s at 0.6 s without applying load. The peak SCs rise from 0.45 A to 0.65 A at 12.5 and 50 Hz, respectively. A similar SR is shown in Fig. 4f. Nevertheless, a change in SCs from 0.85 A to 1.05 A is monitored at 1 Nm load conditions.

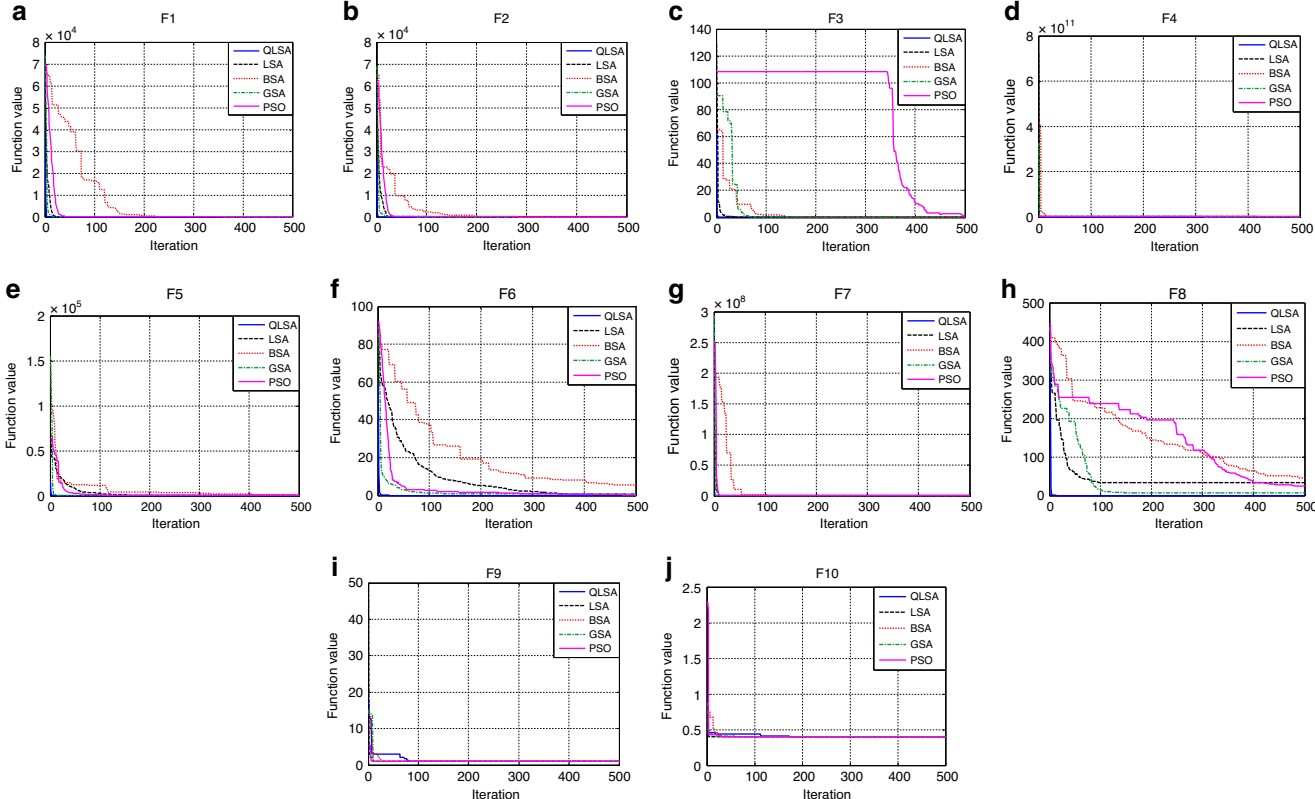

**Fig. 2 The convergence characteristics performance evaluation of QLSA, LSA, BSA, GSA and PSO under different benchmark functions. a** Convergence characteristic curves for QLSA, LSA, BSA, GSA and PSO in benchmark function F1 (Sphere). **b** Convergence characteristic curves for QLSA, LSA, BSA, GSA and PSO in benchmark function F2 (Step). **c** Convergence characteristic curves for QLSA, LSA, BSA, GSA and PSO in benchmark function F3 (Quartic). **d** Convergence characteristic curves for QLSA, LSA, BSA, GSA and PSO in benchmark function F4 (Schwefel 2.22). **e** Convergence characteristic curves for QLSA, LSA, BSA, GSA and PSO in benchmark function F5 (Schwefel 1.2). **f** Convergence characteristic curves for QLSA, LSA, BSA, GSA and PSO in benchmark function F6 (Schwefel 2.21). **g** Convergence characteristic curves for QLSA, LSA, BSA, GSA and PSO in benchmark function F7 (Rosenbrock). **h** Convergence characteristic curves for QLSA, LSA, BSA, GSA and PSO in benchmark function F8 (Rastrigin). **i** Convergence characteristic curves for QLSA, LSA, BSA, GSA and PSO in benchmark function F9 (Foxholes). **j** Convergence characteristic curves for QLSA, LSA, BSA, GSA and PSO in benchmark function F10 (Branin).

Consequently, the MAE, RMSE and SD of QLSAF are 1.6872%, 6.5379% and 6.4662%, respectively (Supplementary Table 7). In summary, QLSAF has enhanced overshoot (OS) and settling time (ST) compared with other methods under different speeds and load operations. Furthermore, QLSAF achieves lower MAE, RMSE and SD than other controllers under different load conditions.

The DTUTD step SR under different load conditions is a challenging test to explore. Figure 4g shows the step DTUTD SR, where the speed is altered from 35 rad/s to 70 rad/s at 0.25 s, from 70 rad/s to 105 rad/s at 0.5 s and from 105 rad/s to 140 rad/s at 0.75 s at no-load condition. Meanwhile, the peak SCs vary at 0.45 A with 12.5 Hz, 0.55 A with 25 Hz, 0.6 A with 37.5 Hz and 0.65 A with 50 Hz. Figure 4h shows SRs that are identical to those in Fig. 4g. However, the peak SCs change at 0.7 A with 12.5 Hz, 0.9 A with 25 Hz, 1 A with 37.5 Hz and 1.05 A with 50 Hz at 1 Nm load condition. The MAE, RMSE and SD achieved by QLSA are 0.7977%, 4.6566% and 4.6430%, respectively (Supplementary Table 8). In Fig. 4i, the DTUTD step SR changes from 70 rad/s to 105 rad/s at 0.3 s, from 105 rad/s to 140 rad/s at 0.6 s and returns to its original speed at 2 Nm condition at 0.9 s. Meanwhile, the peak SCs vary at 0.95 A with 25 Hz, 1.05 A with 37.5 Hz and 1.15 A with 50 Hz. Accordingly, QLSA obtains low steady state error, indicating that the MAE, RMSE and SD values are 1.1879%, 7.7192% and 7.6507%, respectively (Supplementary Table 8). In all cases, the SCs change with the variation of speed and load

under the identical SR. The proposed QLSAF controller is also observed to be superior to other controllers in terms of achieving low MAE, RMSE and SD under each step change in speed or load.

The capability of the proposed controller is further assessed on the basis of the RS test under different load and speed conditions (Supplementary Fig. 8). The ramp SR increases from a speed of 105 rad/s to 140 rad/s at 0.2 s under the no-load condition and then continues with the same repetition speed changes (Supplementary Fig. 8a). Meanwhile, the gradual change in SCs is observed with respect to frequency. Supplementary Figure 8b shows an SR similar to that in Supplementary Fig. 8a, except that the load is changed to 2 Nm. The MAE, RMSE and SD are 3.5827%, 17.4315% and 17.0585%, respectively (Supplementary Table 9). The RS alters from a speed of 70 rad/s to 140 rad/s under the no-load condition and then continues at the same repetition speed change (Supplementary Fig. 8c). In the meantime, the gradual change in SC is observed with the change in frequency. The shape of supplementary Fig. 8(c, d) is relatively similar in terms of ramp SR. Nevertheless, the SCs operate under 2 Nm load. Accordingly, QLSA achieves MAE, RMSE and SD of 2.0831%, 11.1266% and 10.9286%, respectively (Supplementary Table 9). The change in ramp speed is executed from 35 rad/s to 140 rad/s under no load and 1 Nm load conditions (Supplementary Fig. 8e, f). Although the change in ramp SR remains unchanged without load and at 1 Nm load condition, the SCs increase with the application of load. Generally, the QLSAF

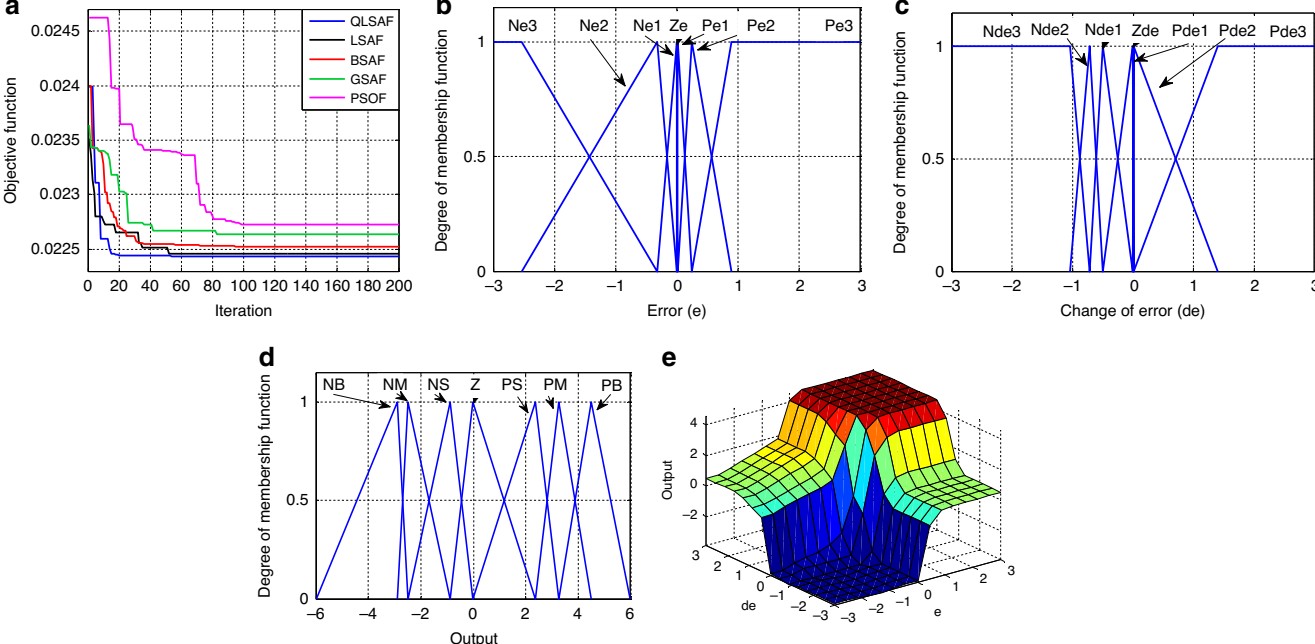

**Fig. 3 The optimization results of QLSA. a** The objective function assessment results of QLSA, LSA, BSA, GSA and PSO in convergence characteristics curve. **b** Optimized membership function for error. **c** Optimized membership function for change of error. **d** Optimized membership function for output. **e** Three-dimensional relationship among error, change of error and output.

controller is superior to other controllers with respect to OS and ST under different cases of ramp speed changes. In addition, QLSAF has lower MAE, RMSE and SD compared with other controllers under changing speed and load conditions. The effectiveness and robustness of the proposed QLSAF in comparison to PID controller is evaluated under two experiments including constant torque with speed variation and constant speed with torque variation (Supplementary Note 5, Supplementary Figs. 10–13 and Supplementary Tables 10, 11).

**Experimental results of DSP-based QLSAF speed controller.** The accuracy and effectiveness QLSAF speed controller are validated under the experimental environment using similar tests executed in MATLAB/Simulink. The experimental tests resulting in the step SRs with varying speeds and load conditions are depicted in Fig. 5. The KEYSIGHT DSO-X2024A oscilloscope is used to monitor the experimental results using four channels for 1 s/Div. The a, b and c phases of SCs are observed using three channels with 200 mA/Div, and the SR is recorded using the fourth channel with 50 mV/Div. The experimental test results are nearly aligned with the simulation results. Figure 5a demonstrates that the SR of the motor accelerates from 105 rad/s to 140 rad/s at 3 s under no-load condition. Subsequently, the motor speed returns to 105 rad/s from 140 rad/s at 6 s, and the peak SCs vary from 0.6 A to 0.65 A at 37.5 Hz and 50 Hz, respectively. This result proves that a proportional relationship exists between speed and frequency. A similar type of SR is shown in Fig. 5b, but the change in SCs is reported from 1.05 A to 1.15 A because of the execution of the 2 Nm load. Figure 5c illustrates the change in speed under the no-load condition from 70 rad/s to 140 rad/s at 3 s and then drops to 70 rad/s at 6 s without OS. The peak SCs vary from 0.55 A to 0.65 A at 25 and 50 Hz. Figure 5c, d are similar with respect to SRs, except for the peak SCs, which increase from 0.95 A to 1.15 A under 2 Nm load situation. Figure 5 shows the increase in speed from 35 rad/s to 140 rad/s at 3 s and then decrease to 35 rad/s at 6 s without any OS under no-load condition. At this time, the peak SCs also increase from 0.45 A to

0.65 A with 12.5 and 50 Hz. The changes in SR in Fig. 5d, f are analogous. Nonetheless, the increment in peak SCs is from 0.7 A to 1.05 A because of the 1 Nm loading.

The DTUTD step speed tests are also implemented in the experimental tests. The oscilloscope image for the experimental results of the DTUTD step response test is shown in Fig. 5g, i. In Fig. 5g, the DTUTD step SR changes at 2.5 s as a step response from 35 rad/s to 70 rad/s, at 5 s from 70 rad/s to 105 rad/s, at 7.5 rad/s from 105 rad/s to 140 rad/s and then return step by step to quarter speed without applying load. Meanwhile, the SCs change with the variation of speed and frequency. Figure 5h shows the same SR as that of Fig. 5g. However, the SCs vary due to the change in frequency and the implementation of 1 Nm load to the TIM. In Fig. 5i, the DTUTD step SR changes from 70 rad/s to 105 rad/s at 3 s, from 105 rad/s to 140 rad/s at 6 s and then returns step by step to half speed with 2 Nm load. In the meantime, the frequency of the SCs changes with the variation of speed and increment of loading on the TIM.

The experiments are also conducted on the basis of RS tests (Supplementary Fig. 9). The experimental reports match the simulation outcomes. Nevertheless, the results are recorded in the time scale of 1 s/Div. The ramp SR changes from 105 rad/s to 140 rad/s at 2 s and continues with a similar SR without loading, in which a gradual change in SCs is observed with its frequency (Supplementary Fig. 9a). The variation of ramp SR in Supplementary Fig. 9a, b is similar. However, the increment in the SCs is reported with the execution of 2 Nm load. The ramp SR varies from 70 rad/s to 140 rad/s at 2 s and continues with a similar result without loading, in which the SCs change increasingly with frequency (Supplementary Fig. 9c). The RSs in supplementary Fig. 9c, d are identical. However, the SC increases under a 2 Nm load. Similar results are also noted in supplementary Fig. 9e, f, in which ramp speed changes from 35 rad/s to 140 rad/s at no-load and 1 Nm load conditions, respectively. Thus, the TIM operated under no-load and load conditions does not affect the ramp SR. Nevertheless, SCs differ with the load increment and change of ramp speed. The experimental results (Supplementary Fig. 9)

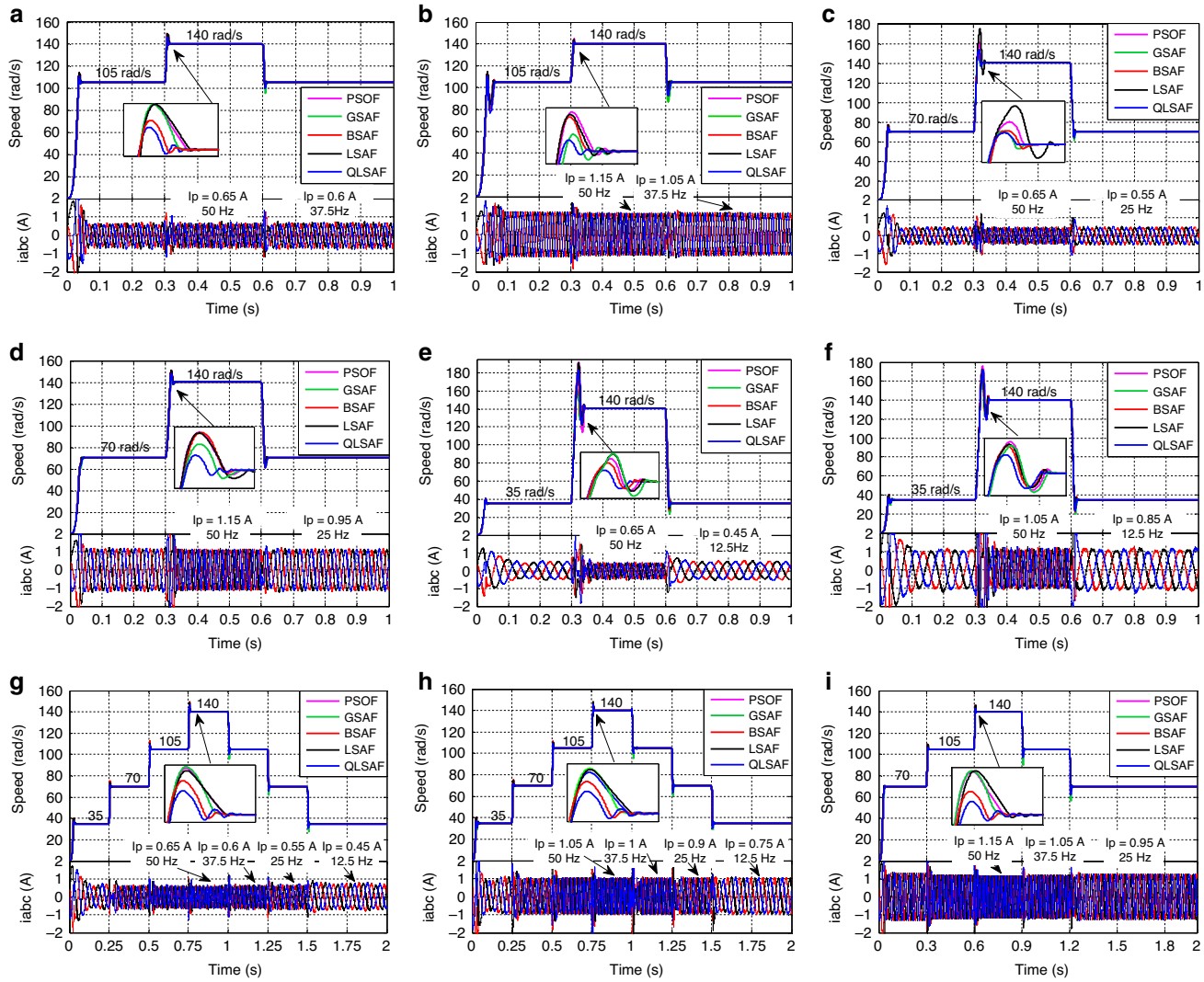

**Fig. 4 Simulation results under step response test and down-to-up-to-down test. a** QLSA performance in step response test under the speed varying from 105 rad/s to 140 rad/s with no-load. **b** QLSA performance in step response test under the speed varying from 105 rad/s to 140 rad/s with 2 Nm load. **c** QLSA performance in step response test under the speed varying from 70 rad/s to 140 rad/s with no-load. **d** QLSA performance in step response test under the speed varying from 70 rad/s to 140 rad/s with 2 Nm load. **e**. QLSA performance in step response test under the speed varying from 35 rad/s to 140 rad/s with no-load and, **f**. QLSA performance in step response test under the speed varying from 35 rad/s to 140 rad/s with 1 Nm load. **g** QLSA performance in down-to-up-to-down test under the speed varying from 35 rad/s to 70 rad/s, from 70 rad/s to 105 rad/s and from 105 rad/s to 140 rad/s at no-load. **h** QLSA performance in down-to-up-to-down test under the speed varying from 35 rad/s to 70 rad/s, from 70 rad/s to 105 rad/s, from 105 rad/s to 140 rad/s at 1 Nm load. **i** QLSA performance in down-to-up-to-down test under the speed varying from 70 rad/s to 105 rad/s, from 105 rad/s to 140 rad/s at 2 Nm load.

under the ramp SRs are consistent with the simulation results (Supplementary Fig. 8), thereby validating the satisfactory solution using the QLSAF speed controller.

## Discussion

An advanced optimisation technique called QLSA is designed to address the optimisation problems of the controller in TIM drive. In addition, an improved FLC controller has been developed to control the TIM drive using the QLSA algorithm. The QLSA controller is implemented on the DSP-TMS320F28335 control board to carry out the validation processes.

The first contribution of this research is the establishment and assessment of the reliability and efficiency of QLSA using 14 benchmark functions with different characteristics. The comparative validation is performed between QLSA and other notable

optimisation techniques, such as the LSA, BSA, GSA and PSO algorithms. The results indicate that the developed QLSA delivers excellent solutions in comparison with LSA, BSA, GSA and PSO algorithms in terms of exploration, exploitation capability and convergence speed.

The second contribution reveals that the design of the QLSA-based FSC achieves high performance in TIM. The optimal control of TIM is achieved by designing the input and output MFs of the FSC with the lowest value of the OF. Hence, the traditional TE method can be avoided. A detailed comparative analysis between QLSAF and other well-known controllers is carried out under changing speed and load environments. The reposts demonstrate that the proposed QLSAF speed controller exhibits superior performance to other controllers with regard to robustness, reduction of damping and improvement of transient responses.

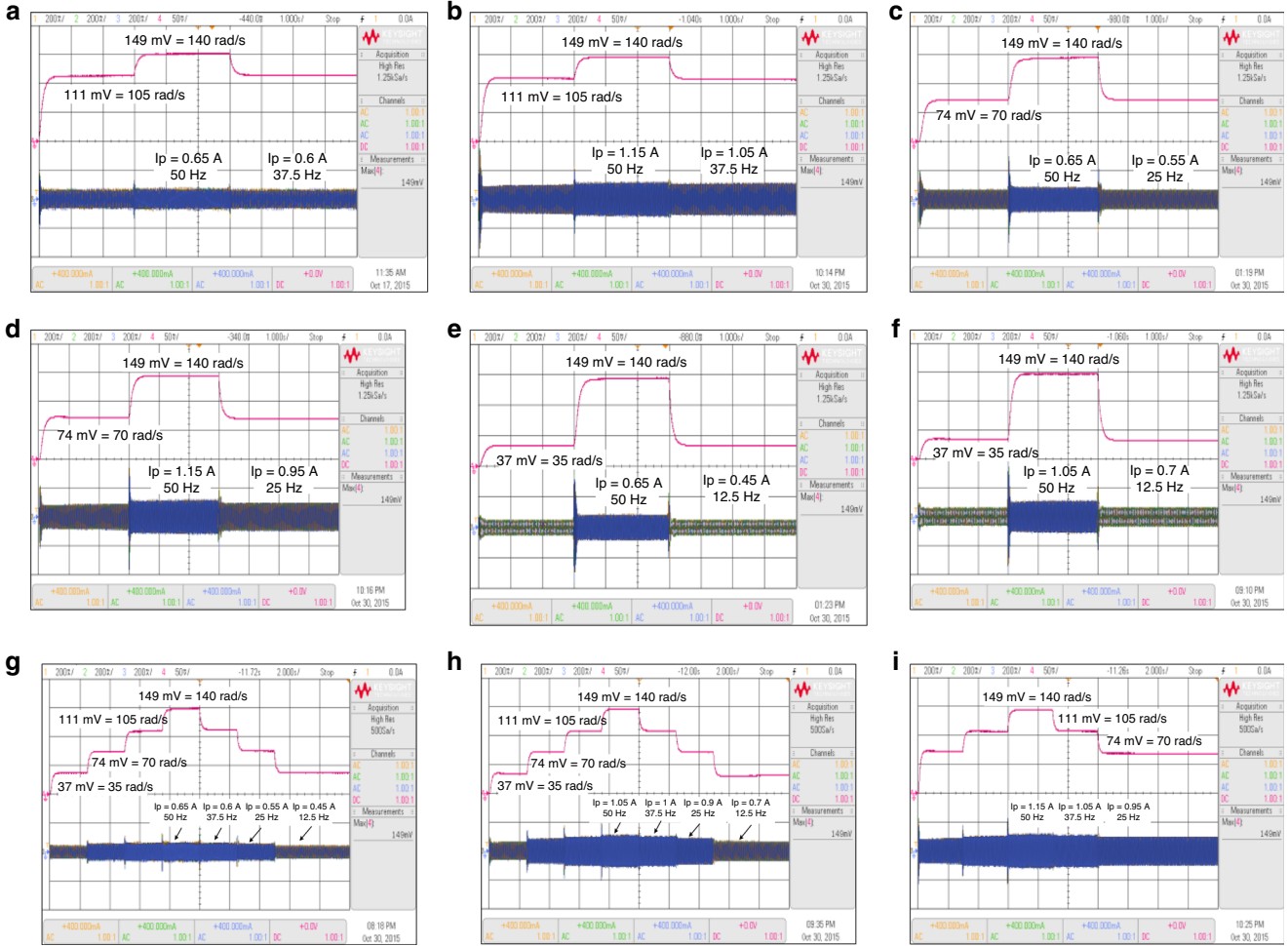

**Fig. 5 Experimental results under step response test and down-to-up-to-down test. a** QLSA performance in step response test under the speed varying from 105 rad/s to 140 rad/s with no-load. **b** QLSA performance in step response test under the speed varying from 105 rad/s to 140 rad/s with 2 Nm load. **c** QLSA performance in step response test under the speed varying from 70 rad/s to 140 rad/s with no-load. **d** QLSA performance in step response test under the speed varying from 70 rad/s to 140 rad/s with 2 Nm load. **e** QLSA performance in step response test under the speed varying from 35 rad/s to 140 rad/s with no-load and, **f** QLSA performance in step response test under the speed varying from 35 rad/s to 140 rad/s with 1 Nm load. **g** QLSA performance in down-to-up-to-down test under the speed varying from 35 rad/s to 70 rad/s, from 70 rad/s to 105 rad/s and from 105 rad/s to 140 rad/s at no-load. **h**. QLSA performance in down-to-up-to-down test under the speed varying from 35 rad/s to 70 rad/s, from 70 rad/s to 105 rad/s, from 105 rad/s to 140 rad/s at 1 Nm load. **i** QLSA performance in down-to-up-to-down test under the speed varying from 70 rad/s to 105 rad/s, from 105 rad/s to 140 rad/s at 2 Nm load.

The third contribution is the implementation of the QLSAF speed control system using a low-cost single-chip DSP-TMS320F28335 control board. The QLSAF speed controller for TIM drive in real-time includes the implementation of analogue-digital conversion, enhanced pulse width modulation, enhanced quadrature encoder pulse (eQEP) and space vector pulse width modulation (SVPWM). Subsequently, the prototype is developed by utilising the DSP-TMS320F28335 controller board. The real-time performance of the inverter behaviour is monitored by developing a graphical user interface programme in code composer studio (CCS) software.

The fourth contribution demonstrates the validation and verification between the simulation and experimental systems. The outcomes under simulation and experimental environments confirm that the proposed QLSAF-based TIM drive system can efficiently handle the changes in the load and speed conditions smoothly. Indeed, the simulation results are better than the experimental results due to the ideal aspects of the simulation. Therefore, the simulation results are perfectly matched with the

experimental results. Therefore, the proposed QLSAF speed controller, with its low-cost prototype, could be a potential candidate for industrial multi induction motor drive systems. It will be interesting to extend this approach to other controllers such as fuzzy type-2 control or hybrid FLC-PID control in the multi-induction motor drive system.

## Methods

**QLSA development process.** LSA[50] is a modern and enhanced optimisation technique, which is designed using the concept of the natural phenomenon of lightning. This research has enhanced the LSA computational capability on the basis of quantum mechanics. We have studied the fundamental principle of LSA and then further improved the searching capability by defining a new position for the population to achieve the best solutions. Global step leaders $(Gsl_{ij}^t)$ of QLSA are initially determined by assessing the average values of the best locations, leading to the lowest value of the assessment. The global minimum and best position of QLSA are achieved through the attraction and convergence of each step leader. The equation for stochastic attractor of step leaders $p_j$ is as follows:

$$p_{ij}^t = \frac{a_{ij}^t \cdot P_{ij,best}^t + b_{ij}^t \cdot Gsl_{ij}^t}{c_{ij}^t \cdot SF},$$
(1)

for $i = 1,2, …, N$, $j = 1,2, …, D$, and $t = 1,2, …, T$, where $N$, $D$ and $T$ represent the population size, the problem dimension and the maximum number of iteration, respectively; $a$, $b$ and $c$ define the random numbers between 0 and 1, which are uniformly distributed; $P^t_{ij,best}$ is the best step leader for each population; $SF$ is the scale factor, which is recommended to assign between 4 and 20. We set the scale factor $SF$ to 10 to execute QLSA.

Each step leader of LSA is assumed to hold a quantum behaviour and its quantum state is expressed by a wave function ($\psi_w$). The probability density function is denoted by $|\psi_w|^2$, which has a potential that is subject to the potential field, where the step leader lies. The centre point of search space in each step leader is located between $P^t_{ij,best}$ and $Gsl^t_{ij}$. The mathematical expression of wave function after $(t + 1)$ iteration is denoted as[51,52],

$$\psi\left(P^{t+1}_{ij}\right) = \frac{1}{\sqrt{L^t_{ij}}} exp\left(-\left|P^t_{ij} - p^t_{ij}\right|/L^t_{ij}\right), \quad (2)$$

where $L^t_{ij}$ stands for the SD of the double exponential distribution, which changes after each iteration number $t$. The double exponential distribution is characterised by the probability density function $Q$, which can be written as follows:

$$Q\left(P^{t+1}_{ij}\right) = \left|\psi\left(P^{t+1}_{ij}\right)\right|^2 = \frac{1}{L^t_{ij}} exp\left(-2\left|P^t_{ij} - p^t_{ij}\right|/L^t_{ij}\right). \quad (3)$$

In turn, the probability distribution function $M_f$ can be formulated as follows:

$$M_f\left(P^{t+1}_{ij}\right) = 1 - \exp\left(-2\left|P^t_{ij} - p^t_{ij}\right|/L^t_{ij}\right). \quad (4)$$

The $j^{th}$ component of position $p_i$ after the iteration $(t + 1)$ can be obtained on the basis of the Monte Carlo method, as expressed in the following equation:

$$P^{t+1}_{ij} = p^t_{ij} \pm \frac{1}{2} L^t_{ij} \ln\left(1/u_{ij}\right), \quad (5)$$

where $\mu_{ij}$ denotes a random number, which is distributed uniformly between 0 and 1. The SD ($L^t_{ij}$) of each step leader is estimated using the following equation:

$$L^t_{ij} = 2\beta\left|Mbest^t_j - P^t_{ij}\right|, \quad (6)$$

where the mean best position for the step leader is represented by $MeanBest^t_j$ and can be defined as the mean value of the $P^t_{ij,best}$ positions of all step leaders. $Mbest^t_j$ of the step leader can be written as follows:

$$MeanBest^t_j = \frac{1}{N} \sum^N_{i=1} P^t_{ij} = \left(\frac{1}{N} \sum^N_{i=1} P^t_{i1}, \frac{1}{N} \sum^N_{i=1} P^t_{i2}, \frac{1}{N} \sum^N_{i=1} P^t_{i3}, … …, \frac{1}{N} \sum^N_{i=1} P^t_{ij}\right). \quad (7)$$

The contraction expansion coefficient ($\beta$) controls the convergence speed of QLSA, which can be written as follows:

$$\beta = \beta_0 + (T - t). \frac{\beta_1 - \beta_0}{T}, \quad (8)$$

where $\beta_0$ and $\beta_1$ represent the initial and final values of the contraction expansion, respectively $t$ and $T$ imply the current and maximum iteration number, respectively. The value of $\beta_1$ is set between 0.8 and 1.2, and $\beta_0$ is set below 0.6 to achieve satisfactory QLSA performance[51]. Therefore, the updated position of step leaders $P^t_{ij}$ can be formulated as follows:

$$P^{t+1}_{ij} = p^t_{ij} \pm \beta\left|MeanBest^t_j - P^t_{ij}\right| \ln\left(1/u_{ij}\right). \quad (9)$$

QLSA has several advanced features compared with the original LSA. Firstly, the QLSA utilises the exponential distribution function to find the new locations between the step leaders through the global convergence. Secondly, the original LSA is enhanced by assessing the mean best position. The new distribution of the step leader is controlled by the distance between step leaders and $MeanBest^t_j$, as expressed in Eq. (9) (Supplementary Fig. 1).

**QLSA verification process**. A group of 14 benchmark functions[50,53,54] was used to validate the accuracy and convergence characteristics of QLSA (Supplementary Table 1). These benchmark functions were characterised into four testing groups. The first group used unimodal and separable functions, including Sphere (F1), Step (F2) and Quartic (F3) to check the strength, reliability and strength, respectively. The second group used unimodal and nonseparable functions, including Schwefel 2.22 (F4), Schwefel 1.2 (F5), Schwefel 2.21 (F6) and Rosenbrock (F7), to assess the performance and consistency. The third group utilised multimodal and separable functions, such as Rastrigin (F8), Foxholes (F9) and Branin (F10), to evaluate the dimensionality problems. The fourth group used multimodal and nonseparable high- and low-dimensional benchmark functions, including Ackley (F11), Griewank (F12), Penalised (F13) and Penalised 2 (F14), to verify the exploration and exploitation capability.

The performance of QLSA was compared with four prominent optimisation techniques, namely, LSA[50], BSA[55], GSA[56] and PSO[57] (Supplementary Notes 1–4, and Supplementary Figs. 2–5). In addition, each benchmark function was tested 50 times. All the optimisation algorithms were operated using a population size of 50

and 500 iterations. In LSA, channel time was set to 10. In BSA, the control parameter, $F$, was set to 3. In GSA, the gravitational constant $G_0$ and acceleration $\alpha$ were 100 and 20, respectively. In PSO, the acceleration coefficients $c_1$, $c_2$ and weight factor $w$ were 1.5 and 0.5, respectively.

**Fuzzy logic speed controller using QLSA**. The fuzzy logic speed controller is well-known because of its simplicity and low implementation cost[58–60]. In addition, FLC exhibits strong performance in nonlinear controller systems without designing any mathematical model[61–64]. The fuzzy speed control has many parameters, such as the MF parameters, number of the rule base and number of the MFs[65–67]. The fuzzy speed control can be improved by optimising the value of these parameters. The FSC is designed using various steps[68,69]. The first step is the knowledge-based approach, which is used to select the position of fuzzy MFs and the number of inputs and outputs. In this research, the input data of FLC for the TIM speed control included error ($e$) and change of error ($de$) for rotor speed ($\omega_{rm}$), as presented in the following equations:

$$e(t) = \omega^*_{rm} - \omega_{rm}(t), \quad (10)$$

$$de(t) = e(t) - e(t - 1), \quad (11)$$

The second step characterises the inputs with convenient linguistic value or level; for instance, "big", "medium" or "small". The trapezoidal and triangular MFs re applied to represent the error and change of error for the FSC of MFs. The error $\mu_e(e)$ and change of error $\mu_{de}(de)$ are defined by variables, namely, $(A_0, A_1, A_2)$ and $(B_0, B_1, B_2)$, respectively, which can be written as follows:

$$\mu_e(e) = \begin{cases} \frac{e - A_0}{A_1 - A_0} A_0 \le e < A_1 \\ \frac{e - A_2}{A_1 - A_2} A_1 \le e < A_2 \end{cases} \quad (12)$$

$$\mu_{de}(de) = \begin{cases} \frac{de - B_0}{B_1 - B_0} B_0 \le de < B_1 \\ \frac{de - B_2}{B_1 - B_2} B_1 \le de < B_2 \end{cases} \quad (13)$$

The third stage describes the control rules and linguistic terms of fuzzy logic to make the appropriate decisions for TIM. Generally, the inference systems are structured either using Mamdani method or Takagi–Sugeno method. In this study, Mamdani is applied because of its simple design and structure. The fuzzy rules are established using the if–then linguistic term, and output MFs are determined between the inputs ($e$, $de$) and the output ($\omega_{sl}$). A total of 49 rules are developed for controlling TIM (Supplementary Table 2) and illustrated in the following equations:

Rule 1: If $e$ is 'Ne3' AND d$e$ is 'Nde3' THEN $\omega_{sl}$ is "NB".
Rule 2: If $e$ is 'Ne3' AND d$e$ is 'Nde2' THEN $\omega_{sl}$ is "NB".
⋮
Rule 48: If $e$ is 'Pe2' AND d$e$ is 'Pde3' THEN $\omega_{sl}$ is "PB".
Rule 49: If $e$ is 'Pe3' AND d$e$ is 'Pde3' THEN $\omega_{sl}$ is "PB".

The final step of the FLC is called defuzzification. This process involves the adjustment, generation and control of crisp value in the output MFs. In this research, the centre of gravity is considered to express the crisp values, as shown in the following equation:

$$O_{crisp} = \frac{\sum^n_i w_i.u_i}{\sum^n_i w_i}, \quad (14)$$

where $n$, $u$ and $w$ denote the number of rules, output MFs and weight coefficient, respectively. The minimum values of $\mu_e(e)$ and $\mu_{de}(de)$ are used to define weights, which can be expressed as follows:

$$w_i = \min\left[\mu_e(e), \mu_{de}(de)\right]. \quad (15)$$

**Objective function formulation**. The optimal value of MFs is achieved by assessing the minimum value of the OF which in turn enhances the accuracy and robustness in the FLC output. In this work, the error in TIM was uniformly distributed, thus MAE was selected as the OF to explain the system performance[70]. The MAE function is estimated using the following equation[58],

$$OF = Min\left(MAE = \frac{1}{l} \sum^M_{m=1}\left|\omega^*_{rm} - \omega_{rm}\right|\right), \quad (16)$$

where $l$ is the number of samples, and $\omega^*_{rm}$ and $\omega_{rm}$ are the reference speed and rotor speed, respectively.

**Optimisation limitations for FLC**. The constraints are selected to enforce the limit of MFs in FLC. Thus, the updated values of MFs are always inside the boundary whilst the QLSA attempts to accomplish the preferred OF. For example, the variable of MFs $X^2_{ij}$ should be located between $X^1_{ij}$ and $X^3_{ij}$ to avoid overlapping. To

tackle this issue, the limitations are imposed as depicted in the following equation:

$$X_{ij}^{P-1} < X_{ij}^{P} < X_{ij}^{P+1}. \qquad (17)$$

After the algorithm development, QLSAF is compared with LSAF, BSAF, GSAF and PSOF algorithms using step response test, DTUTD step SR test and RS test with the same population size (30) and iteration numbers (200) to conduct a fair evaluation. In general, MAE, RMSE, and SD are the common statistical error rate terms used for the assessment and verification of controller and optimization algorithms performance. Moreover, the evaluations of uncertainty and statistical sensitivity analysis are calculated by MAE, RMSE, and SD values obtained under defined conditions. This paper introduces an optimal fuzzy algorithm for dealing the uncertainty to obtain the probability distributions for effective uncertainty and statistical sensitivity analysis. Accordingly, the different statistical error rate terms including MAE, RMSE and SD are applied to verify the performance of QLSA. The mathematical expressions of RMSE and SD are as follows[70]:

$$RMSE = \sqrt{\frac{1}{l}\sum_{m=1}^{H} e_m^2} \qquad (18)$$

$$SD = \sqrt{\frac{1}{l}\sum_{m=1}^{H} (e_m - \eta)^2} \qquad (19)$$

where $e_m$ is the error between the reference speed and estimated speed, $l$ is the number of samples, and $\eta$ is the average values of error.

**Experimental setup**. The block diagram of the closed loop scalar control for TIM drive is provided in Supplementary Fig. 14. Five different capacities of TIMs are used in this experiment (Supplementary Fig. 15). However, we have only shown the result analysis and discussion for the Motor 1 (0.5 HP) because of space limitation. The DSP control programme was written in MATLAB/Simulink in real time and interfaced with CCS at 1 μs sampling time. In the beginning, the proposed controller sent voltage to the selector motor circuit of the connected TIM. Subsequently, the DSP received the measured DC SC of the connected TIM. The QLASF speed controller then determined the optimal MF parameters for the connected TIM. The DSP control programme also received the actual rotor position through eQEP and converted this position into rotor speed. The QLASF speed controller allowed the actual speed to track the reference speed. The feedback signal of the rotor speed was provided by the controller to generate the required inverter frequency that drives the TIM. V/f control was fixed to generate a peak voltage and the required frequency for that speed. Then, the SVPWM technique received two input voltages (i.e. $V_\alpha$ and $V_\beta$) and generated switching signals for the inverter insulated-gate bipolar transistor (IGBTs) to facilitate the smooth operation of the TIM drive. The C-code was generated automatically by MATLAB/Simulink interfaced by the CCS. Then, this code was built in the DSP-TMS320F28335 chip and generated the appropriate switching for the IGBTs. The DSP was connected through eQEP and the actual rotor speed of the TIM was monitored by the rotary encoder. The DSP generated the six PWM signals and then transferred these signals to the IGBTs through gate drives, delivering the required power to operate the TIM.

## Data availability

The data that support the findings of this study are available from the corresponding authors upon reasonable request.

## Code availability

The software code and the examined cases that validated our method are available from the corresponding authors upon reasonable request.

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

## Acknowledgements

This work was supported by the LRGS project grant number 20190101LRGS from the Ministry of Higher Education, Malaysia and bold project 10436494/B/2019093 both under Universiti Tenaga Nasional. The grant number DIP-2018-020 under the Universiti Kebangsaan Malaysia. We also would like to acknowledge the support by the Centre for Green Technology, University of Technology Sydney under grant 321740.2232397.

## Author contributions

M.A.H., A.M. and J.A.A. designed the research. J.A.A., M.A.H. and M.S.H.L. conducted the machine learning optimization and modelling, performed the experiments, analysed the data and wrote the manuscript. K.P.J., T.M.I.M., M.M., A.H., K.M.M. and Z.Y.D. provided study oversight and edited the manuscript. All authors discussed the results and commented on the manuscript.

## Competing interests

The authors declare no competing interests.
