## [Peer Review File · Nature Communications]

Reviewers' Comments:

Reviewer #1:

Remarks to the Author:

I read this paper with a lot of interest. Intelligent control is a fascinating subject and this paper presents a new optimization technique that may be used to improve the fuzzy logic controller of three-phase induction motors. I like the description of the fuzzy controller and the steps of the fuzzy system. The paper has merit and is technically sound. One of the strong points of the paper is its plethora of results. However, there are some limitations that need to be addressed:

- The title is a little bit off the content. It talks about machine performance but it is only tested in TIM. Furthermore, the authors discuss the "role of optimization algorithms" but in fact this paper highlights the QLSAF algorithm. In addition, the paper content is mainly for optimizing a fuzzy logic controller, something that is not reflected in the title. Therefore, the title does not fully express the paper content.
- The authors benchmark the QSLAF algorithm against several optimization algorithms like PSO, LSA, BSA, and GSA. I like this approach but the problem is that I'd like to see a comparison against another type of controller (perhaps PID). The authors state that the fuzzy controller performs better than PID but this also has to be shown in practice. That would add extra merit to the paper.
- I strongly suggest that the authors add a block diagram of the QSLAF algorithm or its pseudocode. This way it would facilitate the reader to understand it and reproduce it.
- The test dataset is not described and no details are given. Therefore, the reader cannot understand the difficulty of the testing problem. In addition, a block diagram of the TIM is useful to the reader.
- In addition, it is not clear the direction of the paper. Do the authors propose a new optimization algorithm or do they propose a new control algorithm?
- It is not clear how the number of the fuzzy rules is determined. How many rules are utilized for controlling of TIM and how are these rules are defined? Why 49?
- How is the number of MB is determined? In other words, how is the resolution of the fuzzy system determined?
- In addition, fuzzy controllers have been widely used together with neural networks to provide the so called neurofuzzy controllers that have been widely applied in several domains and seem to be efficient. I wonder whether the authors are able to test against a neurofuzzy controller, and if not, why?
- Constraint (17) has to be further explained. What do the author imply by "overlapping"? Fuzzy sets by default accommodate overlapping of membership functions.
- The authors have to explain the use of three performance measures MAE, RMSE and SD in this text. Why did they adopt all these three measures?
- For better understanding and presentation of the results is to put together the convergence speed of the algorithms in a table.
- A confusing point in the paper is whether the QLSAF is able to capture the global point when optimizing a function. This confusion comes from the evaluation of the algorithm on the 14 functions. This part needs elaboration and it needs to be clarified the conditions under which the algorithm identifies global optimum.
- Lastly, I would like to ask and this is something that the authors have to clarify whether the QLSAF algorithm can be efficient for other type of controllers or only for fuzzy? If yes, the paper contributions is very narrow.

The work of this manuscript is interesting and I am looking forward to seeing the above comments addressed before I raise further concerns. However, I need to emphasize it that the authors need to specify whether this is an optimization or control paper.

Reviewer #2:

Remarks to the Author:

The authors of the paper describe their proposed approach for Role of Optimization Algorithms in Achieving Efficient Machine Performance. The topic is interesting and with possible applicability. However, the paper needs several improvements:

- 1) the main contribution and originality should be explained in more detail, optimization of fuzzy controllers?
- 2) the motivation of the approach with needs further clarification
- 3) discussion of related work in optimization of fuzzy control should be expanded with more recent work
- 4) Minor grammar and syntax issues need correction
- 5) more simulation results and formal comparison of results are needed
- 6) the conclusions should be extended with more future work
- 7) More references to optimization of fuzzy control papers should be included, like:

Optimization of fuzzy controller design using a Differential Evolution algorithm with dynamic parameter adaptation based on Type-1 and Interval Type-2 fuzzy systems. *Soft Comput.* 24(1): 193-214 (2020)

Comparative Study in Fuzzy Controller Optimization Using Bee Colony, Differential Evolution, and Harmony Search Algorithms. *Algorithms* 12(1): 9 (2019)

An approach for parameterized shadowed type-2 fuzzy membership functions applied in control applications. *Soft Comput.* 23(11): 3887-3901 (2019)

A generalized type-2 fuzzy logic approach for dynamic parameter adaptation in bee colony optimization applied to fuzzy controller design. *Inf. Sci.* 460-461: 476-496 (2018)

A new fuzzy bee colony optimization with dynamic adaptation of parameters using interval type-2 fuzzy logic for tuning fuzzy controllers. *Soft Comput.* 22(2): 571-594 (2018)

Response to Reviewers

Manuscript ID: NCOMMS-20-11397

Type of Manuscript: Research Article

Title: Role of Optimization Algorithms in Achieving Efficient Machine Performance

Authors: M A Hannan, M S Hossain Lipu, Mahmuda Akhtar, R A Begum, Md. Abdullah Al Mamum, Aini Hussain, Hassan Basri

All the comments raised by the reviewers are addressed and corrections are highlighted in **RED COLOUR** in the revised paper.

Reviewer #1

I read this paper with a lot of interest. Intelligent control is a fascinating subject and this paper presents a new optimization technique that may be used to improve the fuzzy logic controller of three-phase induction motors. I like the description of the fuzzy controller and the steps of the fuzzy system. The paper has merit and is technically sound. One of the strong points of the paper is its plethora of results. However, there are some limitations that need to be addressed:

Comment 1: The title is a little bit off the content. It talks about machine performance but it is only tested in TIM. Furthermore, the authors discuss the “role of optimization algorithms” but in fact this paper highlights the QLSAF algorithm. In addition, the paper content is mainly for optimizing a fuzzy logic controller, something that is not reflected in the title. Therefore, the title does not fully express the paper content.

Authors Response: Thank you for your comments. Based on suggestion, we have revised the title into 2 (two) new title as follows;

1. Role of Optimization Algorithms based Fuzzy Controller in Achieving Induction Motor Performance Enhancement
2. Role of Quantum Lighting Search Algorithm Based Fuzzy Controller toward Induction Motor Performance Enhancement

We have included title no. 1 in the manuscript to highlight the impact of optimized algorithm in induction motor performance and readability. However, if you find that the title no. 2 is more suitable, we will fix the title 2.

Authors action in the manuscript:

Role of Optimization Algorithms based Fuzzy Controller in Achieving Induction Motor Performance Enhancement

Comment 2: The authors benchmark the QSLAF algorithm against several optimization algorithms like PSO, LSA, BSA, and GSA. I like this approach but the problem is that I'd like to see a comparison against another type of controller (perhaps PID). The authors state that the fuzzy controller performs better than PID but this also has to be shown in practice. That would add extra merit to the paper.

Authors Response: Thank you for your comments. The comparison between QLSA based fuzzy controller and PID controller is shown in the supplementary file. We have conducted two experiments (1) constant torque with speed variation and (2) constant speed with torque variation.

Authors action in the manuscript:

Comparison between QLSA and PID controllers

The effectiveness and robustness of the proposed QLSA-FLC in comparison to PID controllers is assessed under two experiments (1) constant torque with speed variation and (2) constant speed with torque variation.

Test 1: Constant Torque with Speed Variation

The first test involves increasing or decreasing the reference speed while maintaining a fixed torque. This case study aims to evaluate the performance of the proposed FLC and to estimate the reference speed variation with the constant torque of the TIM controlled by the V/F ratio. V/F control generally exhibits weak performance in low-speed applications. However, V/F ratio controls 25%–100% of the nominal speed of the TIM. The performance of the developed FLC in terms of reference speed involves step responses. The performance of the TIM drive during step response change is determined under the condition of a constant torque load applied on the TIM rotor shaft. By contrast, the no-load condition is applied to the TIM with variable speed in short periods, as illustrated in Fig. 6. A controller is used to sustain TIM performance. This study proposes a unique robust controller structure to indicate the speed responses of QLSAF, LSAF, BSAF, GSAF, PSOF, and PID with a nearly perfect speed change. The induction motor applies the reference speed change several times, as presented in Table 8. Table 8 shows the speed response, which varies based on specific durations, and the overshoot (%) values. The maximum overshoot is calculated as maximum overshoot (%) = $\frac{N_{overshoot} - N_{rated}}{N_{rated}} \times 100\%$. QLSAF successfully achieves the best result compared to the other optimization algorithms in terms of maximum overshoot values and settling time. QLSAF achieves better responses than that of LSAF, BSAF, GSAF, PSOF, and PID in terms of minimizing overshoot values, settling time, steady-state error, and damping ratio. After each change, QLSAF establishes excellent rapid stability during each speed change. None of these results can be obtained without a perfect controller, such as the one proposed in this study. Fig. 7 shows that the stator current signal during the start-up of the TIM involves a high current pull and, subsequently, stable signals. The changes in the frequency of the peak value are also fixed during the duration of sudden changes in speed based on system requirements. Controlling speed change corresponds to a change in supply frequency.

Fig.6. Speed response for constant load with speed variations in steps.

Table 8. The speed response of QLSAF, LSAF, BSAF, GSAF, PSOF, and PID controllers to changes in the reference speed.

Period (s)	Reference speed (rad/s)	Maximum overshoot (%)					PID control
		QLSAF	LSAF	BSAF	GSAF	PSOF	
0–0.5	157	4.45	7.64	7.00	12.10	12.11	13.15
0.5–0.7	118	1.27	5.08	8.47	2.96	9.32	9.81
0.7–0.9	78	0.64	4.48	7.69	0.71	10.21	10.92
0.9–1.4	39	1.28	7.69	10.2	1.32	12.82	16.67

1.4–1.6	78	2.56	3.84	3.20	10.26	6.41	12.30
1.6–1.8	118	2.11	2.54	2.33	6.77	6.81	8.305
1.8–2	157	2.54	2.86	2.54	7.00	7.21	8.280
Period (s)	Reference speed (rad/s)	Settling time (s)					
		QLSAF	LSAF	BSAF	GSAF	PSOF	PID Control
0–0.5	157	0.054	0.055	0.055	0.06	0.07	0.125
0.5–0.7	118	0.011	0.014	0.015	0.012	0.025	0.042
0.7–0.9	78	0.009	0.012	0.018	0.011	0.026	0.055
0.9–1.4	39	0.009	0.010	0.015	0.011	0.024	0.125
1.4–1.6	78	0.015	0.010	0.010	0.011	0.027	0.056
1.6–1.8	118	0.018	0.014	0.011	0.012	0.022	0.041
1.8–2	157	0.015	0.016	0.012	0.017	0.027	0.044

Fig. 7. Stator currents with the change in the reference speed.

Test 2: Constant Speed with Torque Variation

This test aims to determine system performance and robustness of the proposed controller at full rotor speed with changes in mechanical load is evaluated. The results obtained are shown in Fig. 8, which also displays the speed response and its zoomed locations for each step when the load changes. The estimated speed is considered consistent with the actual speed with good accuracy. In terms of the steady-state error between the reference and actual speeds and damping minimization, QLSAF obtains a better response than LSAF, BSAF, GSAF, PSOF and PID. Fig. 9 presents the stator currents. Constant frequency and variable peak values are modified by step changes in mechanical load for a specific duration. Table 9 lists the mechanical load variations according to specific time durations, settling time and overshoot (%) values. QLSAF achieves the lowest overshoot values among the controllers, and these values allow for inducing the best response.

Fig. 8. Full speed response with mechanical load variations.

Fig. 9. Stator currents with the change in the mechanical load.

Table 9. Detailed speed response of QLSAF, LSAF, BSAF, GSAF, PSOF, and PID controllers to changes in mechanical load

Period (s)	Load (N-m)	Maximum overshoot (%)					Settling time (s)						
		QLSAF	LSAF	BSAF	GSAF	PSOF	PID control	QLSAF	LSAF	BSAF	GSAF	PSOF	PID control
0–0.3	0	4.45	7.64	7.00	12.10	12.11	13.15	0.0540	0.0550	0.0550	0.0600	0.0700	0.125
0.3–0.5	1.247	0.127	0.159	0.191	0.382	0.254	1.464	0.0030	0.0032	0.0031	0.0035	0.0200	0.011
0.5–0.7	2.495	0.127	0.191	0.222	0.350	0.254	1.745	0.0032	0.0032	0.0032	0.003	0.0160	0.032
0.7–0.9	3.742	0.095	0.159	0.159	0.318	0.223	2.292	0.0045	0.0045	0.0045	0.0055	0.0200	0.063
0.9–1.2	4.990	0.159	0.178	0.192	0.286	0.224	2.484	0.0050	0.0050	0.0050	0.0050	0.0180	0.035
1.2–1.4	3.742	0.445	0.477	0.445	0.507	0.477	1.401	0.0020	0.0025	0.0025	0.0030	0.0200	0.042
1.4–1.6	2.495	0.371	0.371	0.445	0.477	0.445	1.783	0.0020	0.0045	0.0045	0.0025	0.0200	0.022
1.6–1.8	1.247	0.382	0.392	0.426	0.509	0.445	2.038	0.0020	0.0055	0.0060	0.006	0.0150	0.023
1.8–2	0	0.392	0.382	0.477	0.509	0.482	2.292	0.0020	0.0050	0.0060	0.005	0.0200	0.021

Comment 3: I strongly suggest that the authors add a block diagram of the QSLAF algorithm or its pseudocode. This way it would facilitate the reader to understand it and reproduce it.

Authors Response: Thank you for your comments. The block diagram of QLSA is presented in the supplementary file (Fig. 1).

Authors action in the manuscript:

Quantum Lightning Search Algorithm

The implementation of QLSA is demonstrated in the flow diagram and there are some sections with the gray shadow which shows the main contributions over the LSA algorithm as described in Fig 1. The gray shadow which is illustrated in the flow diagram is added to improve the LSA computational intelligence by calculating the initial population, β factor and *MeanBest* for each projectile.

Fig. 1. Flow diagram of the proposed QLSA algorithm based optimum fuzzy speed controller design procedure

Comment 4: The test dataset is not described and no details are given. Therefore, the reader cannot understand the difficulty of the testing problem. In addition, a block diagram of the TIM is useful to the reader.

Authors Response: Thank you for your comments. The fourteen benchmark functions were tested using the value of dimension problems, search space and function minimum, as shown in Table 1. We have added the block diagram of the TIM in the supplementary file (Fig. 10).

Fig. 10. Block diagram of the closed-loop of scalar control for TIM drive

Comment 5: In addition, it is not clear the direction of the paper. Do the authors propose a new optimization? algorithm or do they propose a new control algorithm?

Authors Response: Thank you for your comments. We have proposed a novel optimization algorithm called a novel quantum-inspired lightning search algorithm (QLSA) to improve the performance of the Fuzzy speed controller of the induction motor by determining the appropriate values of the membership function.

Comment 6: It is not clear how the number of the fuzzy rules is determined. How many rules are utilized for controlling of TIM and how are these rules are defined? Why 49?

Authors Response: Thank you for your comments. In this research, we have used two input; error, change of error and each input has seven (7) membership functions; NB: Negative big; NM: Negative medium; NS: Negative small; Z: Zero; PS: Positive small; PM: Positive medium; PB: Positive big. Hence, the total 49 (7×7) number of fuzzy rules are utilized to solve the 21 problem dimensions included in 7 MFs for error, change of error, and the output, respectively.

Authors action in the manuscript:

The third stage describes the control rules and linguistic terms of fuzzy logic to make the appropriate decisions for TIM. Generally, the inference systems are structured either using Mamdani method or Takagi–Sugeno method. In this study, Mamdani is applied because of its simple design and structure. The fuzzy rules are established using the if–then linguistic term, and output MFs are determined between the inputs (e, de) and the output (ω_{sl}). A total of 49 rules are developed for controlling TIM as listed in Table 2 and illustrated in the following equations:

- Rule 1: If e is ‘Ne3’ AND de is ‘Nde3’ THEN ω_{sl} is ‘NB’.
- Rule 2: If e is ‘Ne3’ AND de is ‘Nde2’ THEN ω_{sl} is ‘NB’.
- ⋮
- Rule 48: If e is ‘Pe2’ AND de is ‘Pde3’ THEN ω_{sl} is ‘PB’.
- Rule 49: If e is ‘Pe3’ AND de is ‘Pde3’ THEN ω_{sl} is ‘PB’.

Table 2. Fuzzy rules of the induction motor speed controller

$de \backslash e$	e	Ne3	Ne2	Ne1	Ze	Pe1	Pe2	Pe3
Nde3		NB	NB	NB	NB	NM	NS	Z
Nde2		NB	NB	NB	NM	NS	Z	PS
Nde1		NB	NB	NM	NS	Z	PS	PM
Zde		NB	NM	NS	Z	PS	PM	PB
Pde1		NM	NS	Z	PS	PM	PB	PB
Pde2		NS	Z	PS	PM	PB	PB	PB
Pde3		Z	PS	PM	PB	PB	PB	PB

NB: Negative big; NM: Negative medium; NS: Negative small; Z: Zero; PS: Positive small; PM: Positive medium; PB: Positive big.

Comment 7: How is the number of MB is determined? In other words, how is the resolution of the fuzzy system determined?

Authors Response: Thank you for your comments. If we understood MB refer here membership function (MF), then in this research, seven (7) membership functions are determined for each input of error and change of error and 7 MF for output. Therefore, 21 rules of the fuzzy system are determined to solve the 21 problem dimensions included in 7 MFs for error, change of error, and the output, respectively.

Comments 8: In addition, fuzzy controllers have been widely used together with neural networks to provide the so-called neuro-fuzzy controllers that have been widely applied in several domains and seem to be efficient. I wonder whether the authors are able to test against a neuro-fuzzy controller, and if not, why?

Authors Response: Thank you for your comments. The neuro-fuzzy controller is an effective in predicting non-linear and complex systems; however, it takes a long time in data training operation and requires large data storage devices.

Comments 9: The constraint (17) has to be further explained. What do the author imply by “overlapping”? Fuzzy sets by default accommodate overlapping of membership functions.

Authors Response: Thank you for your comments. The “overlapping” in Fuzzy sets implies that the starting point of each fuzzy membership function should not be overlapped during the optimization process.

Comments 10: The authors have to explain the use of three performance measures MAE, RMSE and SD in this text. Why did they adopt all these three measures?

Authors Response: Thank you for your comments. MAE, RMSE, and SD are the common statistical error rate terms used in different literatures for the assessment and verification of controller and optimization algorithms performance. Moreover, the evaluations of uncertainty and statistical sensitivity analysis are calculated by MAE, RMSE, and SD values obtained under defined conditions.

Authors Action in the Manuscript

In general, MAE, RMSE, and SD are the common statistical error rate terms used for the assessment and verification of controller and optimization algorithms performance. Moreover, the evaluations of uncertainty and statistical sensitivity analysis are calculated by MAE, RMSE, and SD values obtained under defined conditions. This paper introduces an optimal fuzzy algorithm for dealing the uncertainty to obtain the probability distributions for effective uncertainty and statistical sensitivity analysis. Accordingly, the different statistical error rate terms including MAE, RMSE and SD are applied to verify the performance of QLSA. The mathematical expressions of RMSE and SD are as follows⁷⁹:

Comments 11: For better understanding and presentation of the results is to put together the convergence speed of the algorithms in a table.

Authors Response: Thank you for your comments. It is noticed from Fig. 2 that the QLSA converges to the lowest objective function value which is much quicker than LSA, BSA, PSO, and GSA algorithms. Thus, it can be concluded that QLSA reaches the optimal solutions faster than LSA, BSA, PSO, and GSA algorithms.

Comments 12: A confusing point in the paper is whether the QLSAF is able to capture the global point when optimizing a function. This confusion comes from the evaluation of the algorithm on the 14 functions. This part needs elaboration and it needs to be clarified the conditions under which the algorithm identifies global optimum.

Authors Response: Thank you for your comments. We have addressed your queries accordingly.

Authors action in the manuscript:

The accuracy of QLSA is nearly adjacent to the global minimum in group 1 benchmark functions for Sphere (F1), Step (F2) and Quartic (F3). The second test is implemented using group 2 benchmark functions, and results indicate that the QLSA reaches the best global minimum for Schwefel 2.22 (F4), Schwefel 1.2 (F5), Schwefel 2.21 (F6) and Rosenbrock (F7). QLSA is also verified under group 3 benchmark functions, where the complexity level of the optimisation problem increases. QLSA reaches the best global minimum for F8 and the near-global minimum for F9 and F10. These results demonstrate the strong computational capacity of QLSA in obtaining any local minimum. The proposed QLSA is tested through the benchmark functions of group 4 (F11, F12, F13 and F14). The results illustrate that the best global minimum for QLSA is found in F11 and F12, and the near-global minimum is achieved in other functions.

Comments 13: Lastly, I would like to ask and this is something that the authors have to clarify whether the QLSAF algorithm can be efficient for other type of controllers or only for fuzzy? If yes, the paper contribution is very narrow.

Authors Response: Thank you for your comments. QLSA is a novel algorithm introduced in this study which is developed from a novel lightning search algorithm (LSA) using quantum mechanics theory to generate a quantum-inspired LSA (QLSA). The QLSA improves the search of each step leader to obtain the best position for a projectile using an exponential distribution through the global convergence and by calculating the mean best position. **Yes, QLSA can be efficiently use in other type of controllers.** QLSA is the advanced form of LSA. We found that LSA works effectively not only in the fuzzy controller but also operates satisfactorily in the PID controller and machine learning algorithms. We have added two prominent articles for your clarifications.

1. Sarker, M. R., Mohamed, R., Saad, M. H. M. & Mohamed, A. DSPACE Controller-based enhanced piezoelectric energy harvesting system using PI-lightning search algorithm. *IEEE Access* **7**, 3610–3626 (2019).
2. Hannan, M. A. *et al.* Toward Enhanced State of Charge Estimation of Lithium-ion Batteries Using Optimized Machine Learning Techniques. *Sci. Rep.* **10**, 4687 (2020).

Reviewer #2

The authors of the paper describe their proposed approach for Role of Optimization Algorithms in Achieving Efficient Machine Performance. The topic is interesting and with possible applicability. However, the paper needs several improvements:

Comment 1: The main contribution and originality should be explained in more detail, optimization of fuzzy controllers?

Authors Response: Thank you for your comments. We have extended the contribution and originality of the paper.

Authors action in the manuscript:

The significant contributions of this research are summarised, as follows:

- i. A novel QLSA is introduced and compared with other optimization techniques by using different benchmark functions. Firstly, this research develops a novel LSA using quantum mechanics theory to generate a quantum-inspired LSA (QLSA). The QLSA improves the search of each step leader to obtain the best position for a projectile using an exponential distribution through the global convergence and by calculating the mean best position. Secondly, the proposed QLSA is applied to a group of fourteen benchmark functions and validated with different tests. The obtained results are QLSA compared with LSA, backtracking search algorithm (BSA), gravitational search algorithm (GSA) and particle swarm optimization (PSO).
- ii. An optimal QLSA-based FLC (QLSAF) speed controller is employed to tune and minimize the OF, thereby improving the TIM performance under different speed and load conditions. The results obtained with the QLSAF are compared with the results obtained with the LSA-based FLC (LSAF), BSA-based FLC (BSAF), GSA-based FLC (GSAF), and PSO-based FLC (PSOF), to validate the performance of the developed speed controller. The results are satisfactory in achieving low steady-state error under numerous speed responses (SRs) and load settings.

- iii. The prototype of the QLSA-based fuzzy (QLSAF) speed controller is implemented in a low-cost single-chip DSP-TMS320F28335 control board. A suitable prototype is designed using DSP-TMS320F28335 control board along with three-phase inverter, related gate driving circuit, rotary encoder connect circuit, motor selector circuit, analogue-digital conversion (ADC), enhanced quadrature encoder pulse (eQEP) and enhanced pulse width modulation (ePWM) blocks. The implementation of the QLSAF speed controller is carried out in V/f control with pulse width modulation (PWM) switching technique and DSP-TMS320F28335 control board.
- iv. The proposed method is validated by experiments, and the results of the simulation and experimental system are consistent with that of the TIM drive system. The results validate and confirm the implementation of the proposed algorithm in a multi-induction motor drive.

Comment 2: The motivation of the approach with needs further clarification.

Authors Response: Thank you for your comments. We have clarified the motivation of our research.

Authors action in the manuscript:

The conventional controller, namely, proportional–integral–derivative (PID) has been widely applied to adjust the main parameters of TIM, including rotor flux, torque, speed, current and voltage^{21,22}. However, PID has negatives in terms of appropriate parameter selection due to the trial-and-error (TE) considerations. The artificial intelligence (AI) based controllers including artificial neural network (ANN) and adaptive neuro-fuzzy inference systems (ANFIS) have been performing satisfactorily in motor applications such as fault identification²³, speed assessment²⁴ and harmonics and torque ripple minimization²⁵. However, the AI-based controllers have drawbacks concerning huge data requirement, long learning and training duration. Fuzzy logic controller (FLC) is extensively utilised in real-time TIM control using adaptive modelling under sudden changes^{26–28}. Furthermore, FLC can operate in highly linear and nonlinear systems without considering any mathematical model^{28,29}. Nevertheless, the accuracy of FLC depends on the suitable design and the optimal number of membership functions (MFs), as well as appropriate fuzzy rule generation³⁰. Generally, a TE procedure is used to determine these variables; however, this procedure causes a substantial delay in control operation³¹.

The execution of TIM drive through the experimental platform is carried out using dSPACE, field-programmable gate array (FPGA), or digital signal processor (DSP). The dSPACE and FPGA have illustrated effectiveness in the implementation of grid-integrated voltage source inverter⁵¹ and five-phase voltage source inverter⁵², respectively. Nevertheless, dSPACE and FPGA have shortcomings in terms of cost and working method that cannot operate on a standalone basis. In contract, DSP offers benefits with regard to cost-effectiveness, low power consumption, fast computational capability, and embedding processor^{53,54} and has been excellent in operating TIM drive⁵⁵ and permanent magnet synchronous motor (PMSM)⁵⁶.

Comment 3: Discussion of related work in optimization of fuzzy control should be expanded with more recent work.

Authors Response: Thank you for your comments. We have added a few established references related to optimization of fuzzy control in induction motor drive.

Authors action in the manuscript:

Ali et al.³⁷ introduced backtracking search algorithm (BSA) based FLC for controlling an induction motor speed, thus avoiding exhaustive traditional TE procedure for obtaining MFs. Ranjani & Murugesan¹⁹ proposed particle swarm optimization (PSO) based FLC to determine the optimal fuzzy parameters for achieving the minimum value of the objective function (OF). Pan et al.³⁸ developed an optimal FLC utilizing genetic algorithm (GA) and PSO through the adjustment of control parameters to minimize the OF. Shareef et al.³⁹ established lightning search algorithm (LSA) based FLC to overcome the TE process in achieving the suitable value of MFs. Mutlag et al.⁴⁰ designed an advanced controller using differential search optimization based FLC to obtain the lowest value of OF and best value of MFs.

37. Ali, J. A., Hannan, M. A., Mohamed, A. & Abdolrasol, M. G. M. Fuzzy logic speed controller optimization approach for induction motor drive using backtracking search algorithm. *Measurement* **78**, 49–62 (2016).
38. Pan, I., Das, S. & Gupta, A. Tuning of an optimal fuzzy PID controller with stochastic algorithms for

- networked control systems with random time delay. *ISA Trans.* **50**, 28–36 (2011).
39. Shareef, H., Mutlag, A. H. & Mohamed, A. A novel approach for fuzzy logic PV inverter controller optimization using lightning search algorithm. *Neurocomputing* **168**, 435–453 (2015).
40. Mutlag, A. H., Mohamed, A. & Shareef, H. A nature-inspired optimization-based optimum fuzzy logic photovoltaic inverter controller utilizing an eZdsp F28335 board. *Energies* **9**, (2016).

Authors action in the manuscript:

Comment 4: Minor grammar and syntax issues need correction.

Authors Response: Thank you for your comments. We have carefully checked the grammar and syntax issues accordingly.

Comment 5: More simulation results and formal comparison of results are needed.

Authors Response: Thank you for your comments. The comparison between QLSA based fuzzy controller and PID controller is shown in the supplementary file. We carried out two experiments (1) constant torque with speed variation and (2) constant speed with torque variation.

Authors action in the manuscript:

Comparison between QLSA and PID controllers

The effectiveness and robustness of the proposed QLSA-FLC in comparison to PID controllers is assessed under two experiments (1) constant torque with speed variation and (2) constant speed with torque variation.

Test 1: Constant Torque with Speed Variation

The first test involves increasing or decreasing the reference speed while maintaining a fixed torque. This case study aims to evaluate the performance of the proposed FLC and to estimate the reference speed variation with the constant torque of the TIM controlled by the V/F ratio. V/F control generally exhibits weak performance in low-speed applications. However, V/F ratio controls 25%–100% of the nominal speed of the TIM. The performance of the developed FLC in terms of reference speed involves step responses. The performance of the TIM drive during step response change is determined under the condition of a constant torque load applied on the TIM rotor shaft. By contrast, the no-load condition is applied to the TIM with variable speed in short periods, as illustrated in Fig. 6. A controller is used to sustain TIM performance. This study proposes a unique robust controller structure to indicate the speed responses of QLSAF, LSAF, BSAF, GSAF, PSOF, and PID with a nearly perfect speed change. The induction motor applies the reference speed change several times, as presented in Table 8. Table 8 shows the speed response, which varies based on specific durations, and the overshoot (%) values. The maximum overshoot is calculated as maximum overshoot (%) = $\left(\frac{N_{overshoot} - N_{rated}}{N_{rated}}\right) \times 100\%$. QLSAF successfully achieves the best result compared to the other optimization algorithms in terms of maximum overshoot values and settling time. QLSAF achieves better responses than that of LSAF, BSAF, GSAF, PSOF and PID in terms of minimizing overshoot values, settling time, steady-state error and damping ratio. After each change, QLSAF establishes excellent rapid stability during each speed change. None of these results can be obtained without a perfect controller, such as the one proposed in this study. Fig. 7 shows that the stator current signal during the start-up of the TIM involves a high current pull and, subsequently, stable signals. The changes in the frequency of the peak value are also fixed during the duration of sudden changes in speed based on system requirements. Controlling speed change corresponds to a change in supply frequency.

Fig.6. Speed response for constant load with speed variations in steps.

Table 8. The speed response of QLSAF, LSAF, BSAF, GSAF, PSOF, and PID controllers to changes in the reference speed.

Period (s)	Reference speed (rad/s)	Maximum overshoot (%)					
		QLSAF	LSAF	BSAF	GSAF	PSOF	PID control
0–0.5	157	4.45	7.64	7.00	12.10	12.11	13.15
0.5–0.7	118	1.27	5.08	8.47	2.96	9.32	9.81
0.7–0.9	78	0.64	4.48	7.69	0.71	10.21	10.92
0.9–1.4	39	1.28	7.69	10.2	1.32	12.82	16.67
1.4–1.6	78	2.56	3.84	3.20	10.26	6.41	12.30
1.6–1.8	118	2.11	2.54	2.33	6.77	6.81	8.305
1.8–2	157	2.54	2.86	2.54	7.00	7.21	8.280

Period (s)	Reference speed (rad/s)	Settling time (s)					
		QLSAF	LSAF	BSAF	GSAF	PSOF	PID Control
0–0.5	157	0.054	0.055	0.055	0.06	0.07	0.125
0.5–0.7	118	0.011	0.014	0.015	0.012	0.025	0.042
0.7–0.9	78	0.009	0.012	0.018	0.011	0.026	0.055
0.9–1.4	39	0.009	0.010	0.015	0.011	0.024	0.125
1.4–1.6	78	0.015	0.010	0.010	0.011	0.027	0.056
1.6–1.8	118	0.018	0.014	0.011	0.012	0.022	0.041
1.8–2	157	0.015	0.016	0.012	0.017	0.027	0.044

Fig. 7. Stator currents with the change in the reference speed.

Test 2: Constant Speed with Torque Variation

This test aims to determine system performance and robustness of the proposed controller at full rotor speed with changes in mechanical load is evaluated. The results obtained are shown in Fig. 8, which also displays the speed response and its zoomed locations for each step when the load changes. The estimated speed is considered consistent with the actual speed with good accuracy. In terms of the steady-state error between the reference and actual speeds and damping minimization, QLSAF obtains a better response than LSAF, BSAF, GSAF, PSOF, and PID. Fig. 9 presents the stator currents. Constant frequency and variable peak values are modified by step changes in mechanical load for a specific duration. Table 9 lists the mechanical load variations according to specific time durations, settling time, and overshoot (%) values. QLSAF achieves the lowest overshoot values among the controllers, and these values allow for inducing the best response.

Fig. 8. Full speed response with mechanical load variations.

Fig. 9. Stator currents with the change in the mechanical load.

Table 9. Detailed speed response of QLSAF, LSAF, BSAF, GSAF, PSOF, and PID controllers to changes in mechanical load

Period (s)	Load (N-m)	Maximum overshoot (%)						Settling time (s)					
		QLSAF	LSAF	BSAF	GSAF	PSOF	PID control	QLSAF	LSAF	BSAF	GSAF	PSOF	PID control
0-0.3	0	4.45	7.64	7.00	12.10	12.11	13.15	0.0540	0.0550	0.0550	0.0600	0.0700	0.125
0.3-0.5	1.247	0.127	0.159	0.191	0.382	0.254	1.464	0.0030	0.0032	0.0031	0.0035	0.0200	0.011
0.5-0.7	2.495	0.127	0.191	0.222	0.350	0.254	1.745	0.0032	0.0032	0.0032	0.003	0.0160	0.032
0.7-0.9	3.742	0.095	0.159	0.159	0.318	0.223	2.292	0.0045	0.0045	0.0045	0.0055	0.0200	0.063
0.9-1.2	4.990	0.159	0.178	0.192	0.286	0.224	2.484	0.0050	0.0050	0.0050	0.0050	0.0180	0.035
1.2-1.4	3.742	0.445	0.477	0.445	0.507	0.477	1.401	0.0020	0.0025	0.0025	0.0030	0.0200	0.042
1.4-1.6	2.495	0.371	0.371	0.445	0.477	0.445	1.783	0.0020	0.0045	0.0045	0.0025	0.0200	0.022
1.6-1.8	1.247	0.382	0.392	0.426	0.509	0.445	2.038	0.0020	0.0055	0.0060	0.006	0.0150	0.023
1.8-2	0	0.392	0.382	0.477	0.509	0.482	2.292	0.0020	0.0050	0.0060	0.005	0.0200	0.021

Comment 6: The conclusions should be extended with more future work.

Authors Response: Thank you for your comments. We have elaborated the conclusions with more future works.

Authors action in the manuscript:

The proposed controller for the TIM drive is implemented in a single chip DSP-TMS320F28335 control board which is considered novel and effective. However, following proposals can be considered for future works and developments:

- i. A method can be developed to convert FLC into a formula or transfer function. This method can be implemented in an extensive range of controllers to reduce the long computational process in FLC.
- ii. New direct torque control can be designed and implemented to control a multi induction motor drive.
- iii. The developed optimization techniques can be applied to other controllers such as fuzzy type-2 control, model-free control or hybrid FLC-PI control, to enhance the control of the multi-induction motor drive system.
- iv. The developed controller can be implemented on a multi DC motor or multi permanent magnet synchronous motor drive to minimize the manufacturing cost of the control system.

Comment 7: More references to optimization of fuzzy control papers should be included, like:

-Optimization of fuzzy controller design using a Differential Evolution algorithm with dynamic parameter adaptation based on Type-1 and Interval Type-2 fuzzy systems. *Soft Comput.* 24(1): 193-214 (2020)

-Comparative Study in Fuzzy Controller Optimization Using Bee Colony, Differential Evolution, and Harmony Search Algorithms. *Algorithms* 12(1): 9 (2019)

-An approach for parameterized shadowed type-2 fuzzy membership functions applied in control applications. *Soft Comput.* 23(11): 3887-3901 (2019)

-A generalized type-2 fuzzy logic approach for dynamic parameter adaptation in bee colony optimization applied to fuzzy controller design. *Inf. Sci.* 460-461: 476-496 (2018)

-A new fuzzy bee colony optimization with dynamic adaptation of parameters using interval type-2 fuzzy logic for tuning fuzzy controllers. *Soft Comput.* 22(2): 571-594 (2018)

Authors Response: Thank you for your comments. We have included the aforementioned references accordingly.

Authors action in the manuscript:

Ochoa et al.⁴¹ deployed Type-1 and Interval Type-2 fuzzy systems to enhance the performance of differential evolution (DE) algorithm to achieve dynamic adaptation of the mutation parameters as well as optimize the MFs. Castillo et al.⁴² analyzed and compared the FLC optimization algorithms including bee colony optimization (BCO), DE, and harmony search algorithms (HSA). Melin et al.⁴³ applied shadowed type-2 fuzzy MFs to reduce the computational cost in control applications. Castillo et al.⁴⁴ optimized the generalized type-2 fuzzy logic system (GT2FLS) with BCO to achieve the optimal configuration of MFs. Angulo and Castillo⁴⁵ presented BCO based FLC to find the optimal design of MFs in complex and non-linear systems.

41. Ochoa, P., Castillo, O. & Soria, J. Optimization of fuzzy controller design using a Differential Evolution algorithm with dynamic parameter adaptation based on Type-1 and Interval Type-2 fuzzy systems. *Soft Comput.* **24**, 193–214 (2020).
42. Castillo, O. *et al.* Comparative Study in Fuzzy Controller Optimization Using Bee Colony, Differential Evolution, and Harmony Search Algorithms. *Algorithms* **12**, 9 (2018).
43. Melin, P., Ontiveros-Robles, E., Gonzalez, C. I., Castro, J. R. & Castillo, O. An approach for parameterized shadowed type-2 fuzzy membership functions applied in control applications. *Soft Comput.* **23**, 3887–3901 (2019).
44. Castillo, O. & Amador-Angulo, L. A generalized type-2 fuzzy logic approach for dynamic parameter adaptation in bee colony optimization applied to fuzzy controller design. *Inf. Sci. (Ny)*. **460–461**, 476–496 (2018).
45. Amador-Angulo, L. & Castillo, O. A new fuzzy bee colony optimization with dynamic adaptation of parameters using interval type-2 fuzzy logic for tuning fuzzy controllers. *Soft Comput.* **22**, 571–594 (2018).

Reviewers' Comments:

Reviewer #1:

Remarks to the Author:

Thanks for addressing my comments. I appreciate your effort and the fact that you recognized the major mistake with the initial title.

Reviewer #2:

Remarks to the Author:

The authors have addressed all my concerns and in my opinion the paper can be accepted.

Reviewers' Comments:

Reviewer #1:

Remarks to the Author:

Thanks for addressing my comments. I appreciate your effort and the fact that you recognized the major mistake with the initial title.

Author response: Thank you for your useful comments.

Reviewer #2:

Remarks to the Author:

The authors have addressed all my concerns and in my opinion the paper can be accepted.

Author response: Thank you for your useful comments.

Two reviewers did not raise any concerns and issues about the paper and their comments are very positive toward the acceptance of the paper.